# The Relationship between Metacognition, Rumination, and Sleep in University Students with a Tendency toward Generalized Anxiety Disorder

**DOI:** 10.3390/bs14060444

**Published:** 2024-05-24

**Authors:** Hui Yu, Zhanbiao Shi, Mei Zhao

**Affiliations:** Institute of Psychology, Chinese Academy of Sciences, University of Chinese Academy of Sciences, Beijing 100101, China; amelie1314@126.com (H.Y.); zhaomei@psych.ac.cn (M.Z.)

**Keywords:** metacognition, rumination, sleep, generalized anxiety disorder

## Abstract

People with generalized anxiety disorder tend to have sleep problems, and studies have found correlations between metacognition, rumination, and sleep, but it is unclear how metacognition and rumination work in people with a tendency towards generalized anxiety disorder. The goal of this paper is to investigate the correlation between metacognition, rumination, and sleep in university students with a tendency towards generalized anxiety disorder, and the mediating role of rumination in the effect of metacognition on sleep. The Generalized Anxiety Disorder Scale (GAD-7), the Meta-Cognition Questionnaire (MCQ-30), the Ruminative Responses Scale (RRS), and the Insomnia Severity Index (ISI) were used to investigate and psychometrically measure 566 university students in Anyang Normal College. The results of correlation analysis showed significant positive correlations between metacognition and sleep, ruminative thinking and sleep, and metacognition and rumination in university students with a tendency towards generalized anxiety disorder. Mediation analysis showed that rumination partially mediated the effect of metacognition on sleep, with the mediating effect accounting for 51.1% of the total effect. There is a strong correlation between metacognition, rumination, and sleep in university students with a tendency towards generalized anxiety disorder, and both metacognition and rumination can predict sleep, while metacognition can affect sleep through the mediating effect of rumination.

## 1. Introduction

According to the 2022 World Health Organization (WHO) report, around 13% (970 million) of the world’s population suffers from a mental disorder, with anxiety disorders accounting for the highest percentage at 31% (301 million), showing that anxiety disorders are one of the most common mental health problems in the world. The prevalence of anxiety disorders is still rising with the accelerated pace of society and the increasing pressure of life, which seriously affects the physical, psychological, and social functioning of patients and imposes a certain burden on both society and families. Therefore, exploring issues related to anxiety disorders, such as sleep, is important for understanding anxiety disorders and alleviating anxiety-related symptoms.

Generalized Anxiety Disorder (GAD) is the most common anxiety disorder, often manifested as a mood of trepidation and fear without a factual basis or a clear objective object and specific conceptual content, accompanied by phytoneurological symptoms (e.g., dizziness, chest tightness, panic, dyspnea, dry mouth, urinary difficulty, anorexia and nausea, constipation, etc.) and muscle tension, as well as motor restlessness, which not only brings great mental pain and physical discomfort to patients but also seriously affects their quality of life and social functioning [1].

Most patients with anxiety disorders have obvious sleep disorders [2] and their quality of sleep will be extremely problematic, such as difficulty in falling asleep and waking up early. It is also difficult for the patients to fall asleep in a short period of time after waking up and they tend to rely excessively on sleeping medicines and usually wake up in the morning in a depressed mood, which is closely related to the patient’s anxiety. As it is difficult for patients to fall asleep after waking up, it has a greater impact on anxiety. So, the longer the patients are awake at night, the more serious the patients’ anxiety is. It was also found that there is a close correlation between the sleep status of patients with anxiety disorders and patients’ anxiety in clinical studies [3,4]. Moreover, anxiety symptoms and sleep disorders interact with each other, aggravating their impact on patients’ lives, studies, and work, thus reducing patients’ quality of life [5].

Metacognition is defined as the ability to reflect on one’s mental state and govern their thoughts and beliefs [6]. The conceptual model of metacognition was proposed by Wells in 1995. Wells used a metacognitive model to explain Generalized Anxiety Disorder; in this model, he categorized worries into two different types: Type 1 and Type 2. Type 1 focuses on external issues and internal non-cognitive issues such as social and health issues and is a normal healthy worry. Type 2 is concerned with worrying about one’s own cognitive events and cognitive processes, i.e., worrying about worry. Jn patients with generalized anxiety disorder, patients are in a state of cognitive dissonance where they have both positive and negative beliefs about worry, and the positive beliefs then cause them to use worry as a strategy to deal with their problems because they see that worrying brings benefits to their problem solving. However, prolonged use of this strategy causes them to feel out of control, which in turn creates worry about worrying, and a descent into generalized worrying. In this process, the more they try to avoid and control this worry, the more it makes them feel out of control, which exacerbates the process, and affects the individual’s appraisal of his or her self-psychological state. At the same time, as the degree of meta-anxiety deepens, it penetrates into the level of consciousness (metacognitive awareness), which further deepens the meta-anxiety, and subsequently traps the individual in a vicious cycle of meta-anxiety, leading to more severe meta-anxiety [7,8].

Existing studies have shown that metacognition plays a crucial role in sleep [9] and that inappropriate metacognitive activities related to sleep may be a contributing factor to sleep difficulties and insomnia [10]. As a feature of insomnia disorder, metacognitive dysfunctions play a crucial role in its genesis and maintenance as well as in modulating sleep reactivity [11]. In addition, sleep-related metacognitive beliefs have effects on the perception of sleep difficulties in older adults [12] and eveningness showed small correlations with having less cognitive confidence and metacognitive beliefs about uncontrollable thoughts [13].

In a study on patients with anxiety disorders, it was shown that metacognitions, worry, and sleep were significantly correlated. It showed a unidirectional relationship between negative metacognitions and prolonged worry processes and a bidirectional relationship between worry and sleep quality [14]. Another non-anxiety disorder-related paper showed that pre-sleep worrying strategies play a mediating role in the relationship between metacognition and sleep [15].

Although the relationship between sleep-related metacognition and poor sleep quality is well recognized, the possible contribution of general metacognitive functioning is still unclear, especially among patients with generalized anxiety disorders.

Ruminative thinking has been explained as a non-adaptive way of reacting [16], which refers to an individual’s tendency to spontaneously think about a negative event over and over again after experiencing a stressful event [17,18]. This way of reacting keeps the individual focused on the negative effects of the negative life event on him or her, thus perpetuating the individual’s experience of negative emotions. The increase in negative emotions in turn increases the risk of individuals suffering from sleep disorders [19].

Research suggests that rumination is also related to sleep difficulties, poor sleep quality, short sleep duration, and delayed sleep duration [20,21]. Other studies have found that ruminative thinking can present a mediating role between other factors and sleep, such as the effect of loneliness [22], negative emotions [23], and stress and stressful life events [24,25] on sleep quality. Also, rumination presents a mediating role in metacognition [26,27].

It is suggested that positive beliefs about rumination initiate rumination in depression [28,29,30,31], and the treatment of metacognition has been found to be effective in improving rumination in patients with social anxiety disorder [32]. Importantly, ruminations (16.6%) emerged as mediators of the relationship between dysfunctional beliefs and attitudes toward sleep and insomnia [33].

In summary, it can be found that there is a significant correlation between metacognition and sleep, rumination and sleep, and metacognition and rumination. However, there are few studies exploring the relationship between metacognition, rumination, and sleep, and no research has involved all three in patients with anxiety disorders.

This study aims to investigate the correlation between metacognition, rumination, and sleep in university students with a tendency toward generalized anxiety disorder and investigate the mediating role of rumination in the effect of metacognition on sleep. The students with a tendency toward generalized anxiety disorder represent those who have symptoms of anxiety along with the possibility of having generalized anxiety disorder and who will receive further screening for anxiety disorders in the next study.

Through this study, the research on the correlation between metacognition, rumination, and sleep can be theoretically added, especially for people with anxiety disorders or a predisposition to anxiety disorders. Moreover, this study can help to better understand anxiety and sleep problems in these people, which can help to provide some direction and rationale for treating anxiety disorders and alleviating their sleep problems.

Based on the evidence above, this study proposes the hypothesis that there is a significant correlation between metacognition, rumination, and sleep among university students with a tendency towards generalized anxiety disorder, and that metacognition has effects on sleep through the mediation of rumination.

## 2. Materials and Methods

### 2.1. Participants and Procedure

To investigate the interplay among metacognition, rumination, and sleep patterns in individuals prone to generalized anxiety disorder, an online survey targeting freshmen at Anyang Normal University was conducted. Facilitated by the head of the university’s psychology department, the survey was introduced to the freshmen following the completion of the questionnaire’s design. The department head explained the research objectives before disseminating the online questionnaire for completion. Although primarily aimed at freshmen, the survey unexpectedly attracted some participation from students across other years, all of whom were subsequently included in the study.

In October 2023, questionnaires were distributed, and subsequently, 8000 responses were collected. Among those, the test results of individuals with GAD-7 scores of 10 and above were screened, leading to the identification of 572 students exhibiting tendencies towards generalized anxiety disorder. After discarding 6 invalid responses, the study focused on 566 students. This cohort comprised 265 males (46.8%) and 301 females (53.2%), with ages ranging from 16 to 24 years.

Later, analysis was conducted on the data gleaned from these 566 participants, aiming to shed light on how their metacognitive processes, rumination habits, and sleep interact in the context of generalized anxiety disorder tendencies. 

### 2.2. Instruments

#### 2.2.1. Generalized Anxiety Disorder Scale (GAD-7, Spitzer RL et al. [34]

A brief self-rating scale for anxiety symptoms developed by Spitzer RL et al. [34] was based on the diagnostic criteria of the American Diagnostic and Statistical Manual of Mental Disorders, 4th edition (DSM-IV).

The GAD-7 has 7 entries, e.g., “Feeling nervous, anxious or eager”; each entry is rated on a 4-point scale from 0 to 3 and the total score is the main statistical index, whereby the higher the score, the worse the degree of anxiety. A total score of 0 to 4 is considered no anxiety or anxiety without clinical significance; 5 to 9 is considered mild anxiety; 10 to 14 is considered moderate anxiety; ≥15 is considered severe anxiety, and the sensitivity and specificity are greatest when the score is taken as 10 [35]. Therefore, in this study, a total score of ≥10 was defined as a positive GAD-7 result, which was recognized as having anxiety; <10 was defined as a negative GAD-7 result, which was recognized as having no anxiety. Studies have shown that the GAD-7 has high reliability and validity for anxiety screening [36,37,38]. Cronbach’s alpha for this scale is 0.928.

#### 2.2.2. Meta-Cognition Questionnaire (MCQ-30, Adrian Wells)

Wells (1997) developed the Meta-Cognition Questionnaire (MCQ) in the field of clinical psychology from a metacognitive perspective for mood disorders, especially anxiety. This questionnaire is a simplified version of the original Meta-Cognition Questionnaire (MCQ-65), which was revised by Wells and Cartwright-Hatton (2004), and despite the streamlining of the questions, it still has high reliability and has performed well in clinical applications [39].

The questionnaire examined five aspects of metacognitive content related to pathology: (1) Cognitive Confidence (CC), an evaluation of one’s own cognitive abilities such as memory and attention, e.g., “I have a poor memory”; (2) Positive Beliefs about Worry (POS), the belief that worry can lead to good results and avoid danger, e.g., “Worrying can help me avoid problems that may arise in the future”; (3) Cognitive Self-Consciousness (CSC), the degree of attention paid to one’s own thinking process, e.g., “I will pay close attention to the way I think”; (4) Negative Beliefs about Uncontrollability and Danger of Worry (NEG), the negative beliefs that worries are uncontrollable and dangerous, e.g., “My worries are dangerous to me”; and (5) Need to Control Thoughts (NC), the belief that one should control thoughts, e.g., “If I can’t control my thoughts, I can’t live very well”. The questionnaire consisted of 30 questions and was scored on a 4-point Richter-type scale: 1—not at all agree to 4—strongly agree.

Meta-Cognition has high reliability and has performed well in clinical applications [39]. In this study, Cronbach’s alpha for this scale reached 0.902.

#### 2.2.3. Ruminative Responses Scale (RRS, Susan Nolen-Hoeksema)

This scale consists of 22 items and is divided into three factors: (1) symptom rumination, e.g., “I often feel how lonely I am”; (2) brooding, e.g., “I often wonder what I did to cause this”; and (3) reflective pondering, e.g., “I often analyze recent events to understand why I feel depressed” [40]. It is rated on a scale of 1–4 (1 = never; 2 = sometimes; 3 = frequently; 4 = always), with higher scores indicating a more severe tendency to ruminate. Studies showed that the scale had good reliability and validity [41]. Cronbach’s alpha for this scale is 0.95.

#### 2.2.4. Insomnia Severity Index (ISI, Charles M. Morin)

The Insomnia Severity Index (ISI), developed in 1993 by American psychologist Charles M. Morin., is a self-rating scale used to assess an individual’s insomnia symptoms over the past two weeks.

The scale consists of seven questions on difficulty falling asleep, sleep maintenance disorders, early awakening, sleep satisfaction, impaired daytime functioning, the impact of insomnia on life, and insomnia concerns, with five options for each question, each representing a different level of severity, ranging from “none” to “very severe”, with each option corresponding to one level of severity.

The scale has good reliability and validity [42] and is one of the most widely used insomnia assessment scales in clinical practice today [43].

Cronbach’s alpha for this scale is 0.89.

#### 2.2.5. Statistical Methods

The relationships among metacognition, rumination, and sleep were explored using Spearman’s correlation analysis. To examine the mediating role of rumination between metacognition and sleep, regression analysis was employed. All statistical analyses were performed using IBM SPSS Statistics 26.0.

## 3. Results

The following figures are based on the 566 persons screened and the screening process and criteria described in Methods.

Table 1 illustrates the demographic composition of the 566 students who qualified for the study. Among them, 265 participants (46.8%) were male and 301 participants (53.2%) were female. The majority, representing 93%, were adults aged between 18 and 24 years, with a minor segment (7%) being under 18 years old. When it came to academic year distribution, there was a predominance of freshmen, who constituted 77% of the participants, compared to 23% from other years. The sample was overwhelmingly from the Han ethnic group, making up 99% of the population. Furthermore, a significant majority (89%) hailed from Henan Province, aligning with the location of their university. In terms of anxiety levels, 63.6% of the 566 students exhibit moderate anxiety levels, 26.9% exhibit moderate-severe anxiety, and 9.5% exhibit severe anxiety.

The distribution of freshmen’s anxiety levels did not differ much from that of non-freshmen, with the percentage of moderate anxiety between 60% and 70%, as indicated in Table 2. However, it is worth noting that freshmen have a higher proportion of moderate anxiety than non-freshmen while non-freshmen have a greater proportion of severe anxiety than first-year students. It appears that there is a rise in student anxiety levels. This could be attributed to non-freshmen being exposed to greater stressors. Alternatively, it could stem from the study primarily focusing on first-year students, rendering the data representative for this group. However, data for non-freshmen are more selective, with fewer participants, thus potentially inaccurately reflecting the anxiety levels within this demographic.

From Table 3, it can be found that among the 566 students with a tendency toward generalized anxiety disorder, more than half (54.95%) of them have problems with sleeping, while 3% of them have serious sleep problems.

### 3.1. Descriptive Analysis and Common Method Bias Test

Table 4 presents the means, minimum, maximum, standard deviations, skewness, and kurtosis of the study variables.

Exploratory factor analysis was conducted using Harman’s one-way test. The results showed that the first common factor had an explanatory rate of 18.86%, which is less than 40%, thus indicating that there is no serious common method bias in this study.

### 3.2. Correlation Analysis

The correlation between the data is analyzed in the figure above (Table 5), which shows that all hypothesized correlations reached significance. Specifically, significant positive correlations were found between sleep and rumination, sleep and metacognition, and metacognition and rumination.

### 3.3. Mediation Analysis

With metacognition as the independent variable, rumination as the mediator variable, and sleep as the dependent variable, a mediation effects test was conducted using Model 4 of the PROCESS macro program in SPSS with 5000 self-sampling tests. Regression analyses, as shown in Table 6, showed a significant total effect of metacognition on sleep, a significant predictive effect of rumination (*p* < 0.001), and a significant predictive effect of rumination on sleep (*p* < 0.001).

Table 7 shows the results of the mediating effect of metacognition on sleep through rumination. The upper and lower bounds of the Bootstrap 95% confidence intervals for the direct effect of metacognition on sleep and the mediating effect of rumination did not contain 0 (see Figure 1), suggesting that metacognition not only directly predicted sleep but also predicted sleep through the mediating effect of rumination, which accounted for 48.9% and 51.1% of the total effect value, respectively. Therefore, rumination plays a partially mediating role in the effect of metacognition on sleep, and the path diagram of the mediating effect is shown in Figure 1.

## 4. Discussion

This study explored the correlations between metacognition, rumination, and sleep in university students with a tendency toward generalized anxiety disorder.

From the results, it is shown that the mean sleep score is 8.78, suggesting that sleep issues are not severe among the participants. However, more than half of the students exhibiting a propensity toward generalized anxiety disorder experience sleep problems, with some even facing severe sleep disturbances. This underscores the importance of not overlooking the sleep problems of these individuals.

Furthermore, significant correlations are observed between metacognition, rumination, and sleep. This indicates that higher levels of metacognition and rumination are associated with more severe sleep problems. Furthermore, similar to the findings of previous studies, our results showed that both metacognition and rumination can significantly affect sleep [11,12,24,44], which suggests that metacognition and rumination can have effects on sleep.

In the present study, we further demonstrated a partial mediating role of rumination in the effect of metacognition on sleep and extended the correlation to people with a tendency toward anxiety disorders. This result suggests that metacognition may indirectly influence sleep problems to some extent by impacting rumination. To explain this, previous studies showed that metacognition can interfere with sleep through intrusive cognitive activities [45] as well as metacognitive beliefs about sleep difficulties [12], whilst patients use thought-control strategies to minimize disturbances to sleep but actually exacerbate sleep difficulties [15]. In this process, rumination associated with metacognitive beliefs may significantly affect an individual’s coping strategy. Generally, positive metacognitive beliefs cause patients to adapt rumination as a coping strategy, whereas negative metacognitive beliefs cause an unadjusted strategy. Combined with this evidence, our results suggested that metacognition can act on sleep through the mediation of rumination. It triggers negative beliefs, which results in negative experiences and negative emotions for the patient [31,46], exacerbating the patient’s symptoms of anxiety [47]. In addition, rumination can trigger overthinking at bedtime, which in turn leads to slow and shallow sleep and also increases an individual’s negative expectations of the future, therefore causing the individual to develop a sense of hopelessness about sleep and form negative expectations and then fall into a negative cycle [48].

Examining the correlation between sleep and the interplay of metacognition and rumination among people with a tendency toward generalized anxiety disorder holds significant research implications. Initially, with anxiety symptoms on the rise in non-clinical populations, understanding sleep disturbances within this demographic is crucial for early intervention and prevention strategies against the development of anxiety symptoms. This research underscores the presence of sleep problems among them, highlighting the importance of addressing these issues with considerable attention. Subsequently, despite the lack of professional treatment, individuals in this group contend with anxiety symptoms that significantly impede their daily functioning and mental well-being. Therefore, investigating sleep issues within this cohort and their nexus with cognitive factors could facilitate the delivery of more comprehensive healthcare services and support for the broader community. Moreover, delving into the roles of metacognition and rumination in this context is pivotal for devising tailored interventions. By discerning the relationship between these cognitive factors and sleep problems, personalized sleep intervention programs can be developed to cater to different cognitive profiles, thereby enhancing intervention effectiveness and sustainability.

This study has some limitations. Firstly, the assessment of sleep quality and anxiety was based entirely on self-rated measures that may return different ratings compared with clinical evaluations. To compensate for this, some clinical tests could be added to the measurement of anxiety and sleep in future studies. Secondly, the subjects in this study were university students with a tendency toward generalized anxiety disorder, and future studies could consider exploring relevant research with people from different groups or with anxiety disorders to better understand anxiety or anxiety disorders. In addition, regarding the effect on sleep, metacognition and rumination are only some of the influencing factors, and other factors can be included to explore in the future.

## 5. Conclusions

There was a correlation among metacognition, rumination and sleep in university students with a tendency towards generalized anxiety disorder, with metacognition significantly predicting the quality of sleep, as well as indirectly predicting sleep quality through the mediating role of rumination, which suggested the important role for interventions of metacognition and rumination in people with a tendency toward anxiety disorders with sleep problems.

## Figures and Tables

**Figure 1 behavsci-14-00444-f001:**
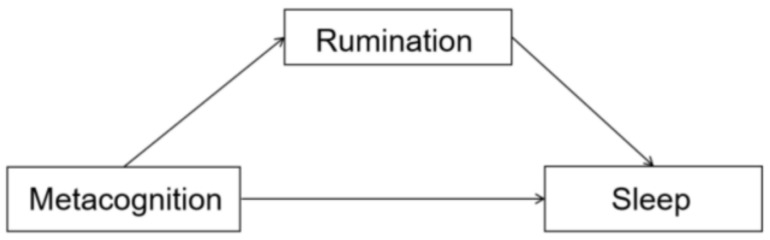
Path diagram of the mediating role of rumination between metacognition and sleep.

**Table 1 behavsci-14-00444-t001:** Demographic characteristics of the study population of the 566 students.

	Demographics	Number	Percentage
Gender	Male	265	46.8%
	Female	301	53.2%
Age (years old)	16–17	39	7%
	18–24	527	93%
Grade	Freshman	436	77%
	Non-freshman	130	23%
Ethnicity	Han	560	99%
	Non-Han	6	1%
Hometown	Henan Province	502	89%
	Non-Henan Provinces	64	11%
Anxiety levels	Moderate anxiety	360	63.6%
	Moderate-severe anxiety	152	26.9%
	Severe anxiety	54	9.5%

**Table 2 behavsci-14-00444-t002:** Anxiety levels among freshmen and non-freshmen.

	Anxiety Levels	Number	Percentage
Freshmen	Moderate anxiety	280	64.20%
	Moderate-severe anxiety	118	27.10%
	Severe anxiety	38	8.70%
	Subtotal	436	100%
Non-freshmen	Moderate anxiety	84	61.50%
	Moderate-severe anxiety	30	26.20%
	Severe anxiety	16	12.30%
	Subtotal	130	100%
Total	Moderate anxiety	360	63.60%
	Moderate-severe anxiety	152	26.90%
	Severe anxiety	54	9.50%
	Subtotal	566	100%

**Table 3 behavsci-14-00444-t003:** Sleep problems among freshmen and non-freshmen.

Sleep Problems	Number	Percentage
Without insomnia	255	45.05%
Mild insomnia	224	39.58%
Moderate insomnia	70	12.37%
Severe insomnia	17	3.00%
Total	566	100%

**Table 4 behavsci-14-00444-t004:** Descriptive analysis of the 566 participants.

Study Variables	Metacognition	Rumination	Sleep
Mean	81.81	59.68	8.78
Minimum	31	22	0
Maximum	120	88	28
Std.Deviation	12.28	13.58	5.81
Skewness	−0.234	0.410	0.634
Kurtosis	2.063	−0.687	0.293

**Table 5 behavsci-14-00444-t005:** Correlations between rumination, metacognition, and sleep.

	Metacognition	Rumination	Sleep
Metacognition	1	0.459 **	0.190 **
Rumination	0.459 **	1	0.254 **
Sleep	0.190 **	0.254 **	1

** Correlation is significant at the 0.01 level (2-tailed).

**Table 6 behavsci-14-00444-t006:** Regression analysis of the relationship between variables in the intermediation model.

Outcome Variables	Predictor Variables	R^2^	F	β	t
Sleep	Metacognition	0.04	21.08	0.19	4.59
Sleep	Rumination	0.07	38.82	0.25	6.23
Rumination	Metacognition	0.21	150.77	0.46	12.28

**Table 7 behavsci-14-00444-t007:** Bootstrap analysis with rumination as a mediating variable.

	Effect	Boot SE	Boot LLCI	Boot ULCI	Effect Ratio (%)
Indirect effect	0.459	0.0115	0.0254	0.0708	51.1
Direct effect	0.440	0.0216	0.0014	0.0428	48.9
Total effect	0.0898	0.0196	0.00001	0.0514	

## Data Availability

The data presented in this study are available upon request from the corresponding author.

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
