# Peer review of "The Relationship between Metacognition, Rumination, and Sleep in University Students with a Tendency toward Generalized Anxiety Disorder"

_behavsci, 2024, doi:10.3390/bs14060444_

Round 1

Reviewer 1 Report

Comments and Suggestions for Authors

The article is interesting and deals with an important topic in a non-clinical sample. Even though the authors provide valuable information, some parts of the article remain unclear. Here are some questions and comments.

- descriptions of the measurements must be standardised, e.g. name of the scale and the original authors of the measurement in brackets, sample items as in the questionnaire on metacognition should be available for each measurement

- it must be explained why the authors used the result 10 to differentiate between students with GAD and without GAD (more info and desctiptive statistics for GAD are needed)

- in the introduction, it would be helpful if the authors differentiated between studies on clinical and non-clinical samples

- if the study was conducted online, how do you know that 8000 questionnaires were distributed - did you collect data from 8000 participants?

- what careful selection was made so that you ended up with 572 participants?

- did you have any information about the participants' previous history of counselling or psychotherapy?

- i think it would be valuable if the authors presented the results of GAD in terms of calculating the severity of GAD - that would be valuable information to see how the severity of GAD is distributed in the non-clinical sample (among students). it is unclear

- although this was not the research problem, but it would be interesting to see if there is a difference between freshmen and non-freshmen, as we know that the first semester is a challenging time for freshmen to adjust to the new life at university.

- Table 2. needs more information about the measured structures, skewness, assimetry, min-max and the title with the number of participants for this data

- it seems to me that the discussion needs more discussion - especially what the practical contribution of the results would be, and if we look at the mean scores of the constructs, we see that the students' scores are below average for sleep, slightly above average for rumination, as well as for metacognition. and also there would be important that authors reflect on their data in the context of GAD in nonclinical sample.

Author Response

Thank you very much for taking the time to review this manuscript. Please find the detailed responses below and the corresponding revisions in track changes in the re-submitted files.

Comments 1: I think it would be valuable if the authors presented the results of GAD in terms of calculating the severity of GAD-that would be valuable information to see how the severity of GAD is distributed in the non-clinical sample(among students).it is unclear

Response 1: Thank you for pointing this out. The results of GAD in terms of calculating the severity of GAD is shown in table 1 in Results (P5) in the revised manuscript. The revisions are marked in red. “In terms of anxiety levels, 63.6% of the 566 students exhibit moderate anxiety levels, 26.9% exhibit moderate-severe anxiety, and 9.5% exhibit severe anxiety. “

Comments 2: although this was not the research problem, but it would be interesting to see if there is a difference between freshmen and non-freshmen, as we know that the first semester is a challenging time for freshmen to adjust to the new life at university.

Response 2: I have differentiated anxiety levels between freshman and non-freshman and this can be found in page 6.

“The distribution of freshmen's anxiety levels did not differ much from that of non-freshmen, with the percentage of moderate anxiety between 60% and 70%, as indicated in Table 2. However, It is worth noting that freshmen have a higher proportion of moderate anxiety than non-freshmen while non-freshmen have a greater proportion of severe anxiety than first-year students. It appears that there is a rise in student anxiety levels. This could be attributed to non-freshmen being exposed to greater stressors. Alternatively, it could stem from the study primarily focusing on first-year students, rendering the data representative for this group. However, data for nonfreshmen is more selective, with fewer participants, thus potentially inaccurately reflecting the anxiety levels within this demographic.”

Comments 3:Table 2. needs more information about the measured structures, skewness, assimetry,min-max and the title with the number of participants for this data

Response 3: Skewness, min-max and the title were added in table 4 in page 6 and 7.

Comments 4: -it seems to me that the discussion needs more discussion-especially what the practical contribution of the results would be, and if we look at the mean scores of the constructs,we see that the students' scores are below average for sleep, slightly above average for rumination,as well as for metacognition. and also there would be important that authors reflect on their data inthe context of GAD in nonclinical sample.

Response 4: The discussion has been revised in page 8 and 9.

Comments 5: descriptions of the measurements must be standardised,e.g. name of the scale and the original authors of the measurement in brackets, sample items as in the questionnaire on metacognition should be available for each measurement

Response 5: Changes have been made in Method (P4&5).

Comments 6:-it must be explained why the authors used the result 10 to differentiate between students with GAD and without GAD (more info and desctiptive statistics for GAD are needed)

Response 6: We used the result of 10 to differentiate between students because we found that studies have shown the GAD-7 scale has the best sensitivity and specificity when a cut-off score of 10 is used. So we believe that this could be used as a screening criterion for further diagnosis of anxiety disorders, i.e., students with a score greater than 10 have a tendency towards generalized anxiety disorder, not just anxiety or symptoms of anxiety. This has already been explained in Instruments and more descriptive statistics has been added in table 1 in results.

Comments 7: -in the introduction, it would be helpful if the authors differentiated between studies on clinical and non-clinical samples

Response 7: Actually, we have differentiated between studies of anxiety disordered samples(clinical) and non-anxiety disordered samples(non-clinical), and for the anxiety disordered samples I specially mentioned “anxiety disorder” or “patients with anxiety disorder, the case is that there are not many studies about the anxiety disordered groups.

Comments 8: -if the study was conducted online, how do you know that 8000 questionnaires were distributed- did you collect data from 8000 participants?

Response 8: The study was introduced to the students by the head of the university's psychology department, and then questionnaires were distributed to them online. And yes, we got data from 8000 participants, including their names, phone numbers and some other personal information.

Comments 9: what careful selection was made so that you ended up with 572 participants?

Response 9: We ended up with those participants whose score of GAD-7 were over 10.

Comments 10:  did you have any information about the participants' previous history of counselling or psychotherapy?

Response 10:Sorry, we didn’t, the information about the participants’ previous history of counselling was not collected.

The other changes have been made in the article in red.

Reviewer 2 Report

Comments and Suggestions for Authors

Thanks to the authors for sharing their manuscript. It seems to me that this is a very interesting study, but some points cause me concern:

1.       I don’t understand why the authors use the term ‘tendency towards generalized anxiety disorder’. Traditionally, GAD-7 scores ≥ are considered indicative of clinically significant anxiety. I recommend that the authors choose some commonly used terms, such as ‘anxiety’ and ‘anxiety symptoms’, or justify the term ‘tendency towards generalized anxiety disorder’ in the introduction.

2.       It would be better if the authors refine the abstract. For example, they can indicate the instruments (at least Generalized Anxiety Disorder-7), the exact sample size (‘566’ instead ‘over 500’), and also remove figures from the abstract showing the strength and direction of correlations.

3.       In Table 2, the authors show the Cronbach’s alpha for all instruments except the Generalized Anxiety Disorder-7. Since this is a key instrument, I would like to see its internal consistency. In addition, the Cronbach’s alpha coefficients are usually described in the Method, rather than in the Results.

4.       I don’t see any information about ethical considerations. Have the authors received permission from the ethics committee? If yes, then you need to specify the name of the ethics committee, the number of the minutes or the date of the meeting. If not, then it should be indicated that the study was carried out in compliance with the principles of the Helsinki Declaration.

5.       The authors used fairly sophisticated statistical analysis methods, but did not indicate whether they calculated the sample size. Please add this information.

6.       Please pay attention to the uniformity of design: the numbers are indicated with or without zero (for example, 0.555, then .555). I also think that it is not necessary to specify numbers in the text that are already indicated in the tables (for example, when describing regression analysis).

Sincerely yours,

the reviewer.

Author Response

Thank you very much for taking the time to review this manuscript. Please find the detailed responses below and the corresponding revisions in track changes in the re-submitted files.

Comments 1: I don't understand why the authors use the term 'tendency towards generalized anxiety disorder'.Traditionally,GAD-7 scores ≥ are considered indicative of clinically significant anxiety.I recommend that the authors choose some commonly used terms, such as 'anxiety' and 'anxiety symptoms',or justify the term tendency towards generalized anxiety disorder'in the introduction.

Response 1: Thank you for pointing this out. We use the term 'tendency towards generalized anxiety disorder' because we are not just choosing those with anxiety symptoms, we are going to further screen for students with anxiety disorders within this group and implement some interventions for them at a later stage. We found that studies have shown the GAD-7 scale has the best sensitivity and specificity when a cut-off score of 10 is used, so we choose result 10 to distinguish.

I added the explanation of the term in the introduction in page 3.

Comments 2: It would be better if the authors refine the abstract. For example, they can indicate the instruments(at least Generalized Anxiety Disorder-7),the exact sample size('566' instead over 500'),and also remove figures from the abstract showing the strength and direction of correlations.

Response 2: The instruments, exact sample size were indicated, and the figures showing the strength were removed in the abstract.

Comments 3: In Table 2, the authors show the Cronbach's alpha for all instruments except the Generalized Anxiety Disorder-7.Since this is a key instrument,I would like to see its internal consistency.

In addition,the Cronbach's alpha coefficients are usually described in the Method,rather than in the Results.

Response 3: The Cronbach's alpha for GAD-7 was shown in the study, and I also moved the Cronbach's alpha for all instruments to Method in page 4.

Comments 4: I don't see any information about ethical considerations.Have the authors received permission from the ethics committee? If yes, then you need to specify the name of the ethics committee,the number of the minutes or the date of the meeting. If not,then it should be indicated that the study was carried out in compliance with the principles of the Helsinki Declaration.

Response 4: This part was added in the paper. The study was approved by the Institutional Review Board of the Institute of Psychology, Chinese Academy of Sciences (H23128, November 10 , 2023).

Comments 5: The authors used fairly sophisticated statistical analysis methods, but did not indicate whether they calculated the sample size.Please add this information.

Response 5: All the results were calculated from the 566 university students and it was described in the first part of the results.

Comments 6: Please pay attention to the uniformity of design: the numbers are indicated with or without zero(for example,0.555, then .555).I also think that it is not necessary to specify numbers in the text that are already indicated in the tables(for example,when describing regression analysis).

Response 6: Thank you for your suggestion, I corrected the uniformity of the numbers and also deleted some numbers that are already indicated in the tables.

More results, statistics and discussion were added in the paper in red.

The response can also be found in the attachment.